# Happiness in University Students: Personal, Familial, and Social Factors: A Cross-Sectional Questionnaire Survey

**DOI:** 10.3390/ijerph19084713

**Published:** 2022-04-13

**Authors:** Yingying Jiang, Chan Lu, Jing Chen, Yufeng Miao, Yuguo Li, Qihong Deng

**Affiliations:** 1School of Energy Science and Engineering, Central South University, Changsha 410083, China; yyjiang@csu.edu.cn (Y.J.); jingchen@csu.edu.cn (J.C.); yufengmiao@csu.edu.cn (Y.M.); 2XiangYa School of Public Health, Central South University, Changsha 410078, China; chanlu@csu.edu.cn; 3Department of Mechanical Engineering, The University of Hong Kong, Hong Kong 999077, China; liyg@hku.hk; 4School of Public Health, Zhengzhou University, Zhengzhou 450001, China

**Keywords:** college, happiness gene, mental health, suicide, teenagers, well-being, young people

## Abstract

Happiness is the foundation of a better life and a goal that people pursue; however, happiness levels among university students are low. The purpose of this study is to explore the main factors influencing student happiness. A nationwide cross-sectional study was conducted in China in 2020. Data on student happiness was collected using the Oxford Happiness Questionnaire, and students’ personal, familial, and social information were obtained using another questionnaire. Logistic regression analysis was employed to examine the association between student happiness and these factors in terms of odds ratio (OR) and 95% confidence interval (CI). A total of 2186 valid questionnaires were obtained. Firstly, student happiness was found to be associated with personal factors. The results found that happiness was significantly associated with state of health, the adjusted OR (95% CI) = 3.41 (2.01–5.79) for healthy students compared to unhealthy students, and that happiness decreased with the student’s age (OR = 0.79 and 95% CI = 0.63–0.98). Secondly, the research suggested that happiness was associated with familial factors. Both frequent contact with family and a harmonious relationship with parents significantly enhanced happiness with ORs (95% CIs) 1.42 (1.17–1.71) and 2.32 (1.83–2.95), respectively. Thirdly, student happiness was associated with several social factors. Students who performed well academically, who went to sleep early, and who were in a loving relationship were found to be happier than those with poor academic performance, went to sleep late, and who were single, for which the ORs (95% CIs) were, respectively, 1.87 (1.51–2.32), 1.50 (1.24–1.81), and 1.32 (1.09–1.60). The survey identified several key personal, familial, and social factors influencing university student happiness, which can provide an effective measure to improve their happiness.

## 1. Introduction

Happiness is a subjective index often used to measure quality of life and refers to individual and social well-being [1]. Many studies have put forward their own views on the definition of happiness [2,3,4,5,6,7]. For example, Veenhoven (1992) indicated that: “*Happiness is the degree to which a person judges whether his quality of life is satisfactory*” [8]. In a word, happiness is a subjective and internal emotion, and requires cognition [9,10]. It affects life expectancy, as happier people tend to have healthier and longer lives [11,12,13]. Therefore, happiness is not only a goal that people pursue, but also a guide and inexhaustible motivation for people’s behavior [14]. A large number of studies have shown that happiness is linked to ideals in many fields, from physical health and mental joyfulness to harmonious interpersonal relationships and career success [15]. According to the recent World Happiness Report, 2019, nearly one in three people are not happy with their lives, with the percentage increasing rapidly over the past decade [16].

Low levels of happiness among university students have been reported worldwide and have received considerable attention [17]. A 2017 survey showed that adults aged 18–25 had the highest rate of depression at 13.1 percent, compared with 7.7 percent for those aged 26–49 and 4.7 percent for those aged 50 and above. In addition, the incidence of suicidal thoughts in young people aged 18–25 was twice as high as in other age groups, and the incidence of suicide was 4.5 times higher than in other age groups [18]. University life is a special period when students start to be independent, but if they are not mature enough to adapt to the changes in their personal lives and studies [19], this may cause students to be easily stressed [20]. A previous study reported that about half the number of college students have moderate stress-related mental health problems [21], and more than 20 percent of college students in China suffer from psychiatric disorders, and this rate has been growing [22,23,24]. According to the 2018 National College Health Assessment Survey, 13 percent of college students have suicidal thoughts and about 2 percent have attempted suicide at least once in the past year [25]. A low level of happiness can have a range of negative consequences: impaired physical and mental health, even severe mental illness or suicide, strained interpersonal relationships, and poor academic performance, all of which will seriously affect their future development and career [25,26]. With increasing numbers of college students worldwide [27], the low level of happiness experienced by students prompts an urgent need to examine its influencing factors so as to be able to take active and effective measures to improve student happiness.

The origins of happiness are very complicated [28]. For a long time, the similarity in happiness levels among family members over generations seems to indicate that genetic factors may play an important role in happiness [29,30]. Some common genes or genetic variants associated with happiness have indeed been found [31,32,33]. However, there is accumulating evidence that both genetic and non-genetic factors play important roles in the origins of happiness [34]. In particular, the rapid decline in happiness cannot be explained by genetic change.

Human beings are group-living creatures with an emotional, thinking consciousness, so relevant non-genetic factors can be divided into individual and collective aspects, and previous studies have also confirmed that they are closely related to happiness. As a force for social development and national prosperity, the healthy growth of college students is of great significance. In addition, although there have been many studies on happiness, most of them focused mainly on adults or teenagers [35]. However, the psychological state of college students is unique as they are at the boundary between adolescence and adulthood, which should be considered separately [36]. Due to the particularity of college students, their collective life can be further divided into family and society. Therefore, this work focuses on the personal, familial, and social factors that affect happiness in university students, and aims to explore some potential variables that may affect happiness.

First of all, happiness is a subjective measure of a person’s well-being and is significantly associated with personal characteristics. A large number of studies have shown that gender makes a difference in happiness, with females generally being happier than males [35,37,38]. Health is also linked to happiness with the rate of happiness being higher in healthy people than in unhealthy people [39,40,41]. For students, those with high academic achievement have a higher level of happiness [42,43].

Second, family relationships also play an important role in happiness, as human beings are high-level creatures with an emotional consciousness; therefore, family is the key place where emotions are cultivated [44]. In particular, the relationship between parents is regarded to be the core of family unity and plays a key role in children’s development [45]. A recent study found that the main sources of children’s happiness are family relationships [46].

Third, people are social, and thus social behaviors are also found to be associated with happiness. Studies have shown that good peer relationships [46], regular physical activity [47,48], regular diet [37,49], and no drug dependence [50] are positively associated with happiness. However, the widespread use of mobile phones, computers, and other electronic devices has led to a substantial increase in internet addiction and insomnia [51], which have been found to be strongly negatively associated with happiness levels [52].

Recently there has been a rapid rise in the suicide rate among young people [53,54] and particularly among college students. The purpose of this study is to explore the main factors influencing student happiness in order to find useful suggestions to improve the quality of their development, and to benefit society and country as well.

## 2. Materials and Methods

### 2.1. Study Protocol and Participants

Between January and June 2020, an online questionnaire survey of happiness in university students was conducted in China. The study protocol was approved by the Ethics Committee of Central South University (Number: XYGW-2020-17). The plan was to conduct surveys in 34 provincial administrative regions, including 23 provinces, 5 autonomous regions, 4 municipalities directly under the central government, and 2 special administrative regions. Each region collected 80 questionnaires, for a total of 2720. Prior to data collection, university students were informed of the nature and objectives of the study. All of the students voluntarily completed an anonymous questionnaire after giving their consent to participate.

In total, 2367 completed questionnaires were received across the country, with a recovery rate of 87.02%. In the questionnaire setting, the current educational background option is included to screen out the questionnaires that do not meet the requirements. In addition, since undergraduate and graduate students are the primary groups in Chinese colleges, it will be more accurate to select college students with this educational background for research, and thus 133 students who did not match those criteria were excluded. Then, 7 students who were of abnormal height (<100 cm or >200 cm) and 6 students of abnormal weight (<30 kg or >200 kg) were excluded. Next, 35 students who did not meet the geographical conditions were excluded, as these conditions may confound the study. Finally, the responses from 2186 valid questionnaires were entered into a database (Figure 1). Of our sample, 52% were male and 48% were female, while 77% were undergraduate students and 23% were graduate students.

### 2.2. Questionnaire Survey

The questionnaire was divided into two parts. See the attached table for the contents of the questionnaire (Table 1). The first 15 short surveys collected basic information about the students, which included personal factors (*sex, age, body mass index (BMI),*
*grade**, discipline, health*), familial factors (*contact with family, relationship between parents*), and social factors (*living expenses, sleeping habits, exam scores, relationship status*) (Table 1). Corrections between these factors were reported in Table 2. 

As for the measurement of well-being, researchers have proposed a variety of ways, which can be divided into single-item indicators and multi-item indicators. The reliability and accuracy of the former have been questioned in the survey process: researchers cannot estimate the internal consistency of a single indicator [55], that is, the same subject will have different degrees of well-being results at an interval of one hour in the same test item. In addition, a single-item index cannot capture the multi-dimensional aspects of psychological structure [56]. Multi-item indicators increase validity through the aggregation of multiple indicators, which has proven to be more reliable in the measurement of well-being.

Furthermore, there are many reliable methods in multi-item measurement, but each method has its own best-use conditions. For example, after consulting the relevant literature of the SWLS questionnaire, it is found that the questionnaire is suitable for different age groups [57]. An online survey was conducted worldwide in 2018, and multiple linear regression analysis was used to study the influencing factors of dentists’ subjective well-being [58]. A cross-sectional observational study conducted in 2019 investigated the relationship between migraine patients and life satisfaction through an online questionnaire [59]. However, this study only looked at college students. After consulting the relevant research on the well-being of college students, it is found that the Oxford well-being questionnaire is in agreement with reality, and is a good metric with which to study the predictive relationship between social media addiction and the well-being of college students [60]. 

Furthermore, the Oxford well-being questionnaire items can be easily included in a larger questionnaire in random order and with reversed items in the questionnaire. These changes can reduce the possibility of context and adaptive response and improve the reliability of data [61]. Moreover, compared with other subjective well-being indicators, SWLS has a weaker correlation with emotion [57]. Therefore, the Oxford Happiness Questionnaire was used to gather data related to happiness. It is a 29-item measure that utilizes a six-point rating scale of agreement, ranging from 1 (strongly disagree) to 6 (strongly agree) (Figure 2).

### 2.3. Statistical Analysis

Data were entered into Microsoft Excel and were analyzed using SPSS version 16.0 statistical software. We used multiple logistic regression models to analyze the data. Happiness was coded as a dichotomous variable with 1 = “happiness” and 0 = “unhappiness” according to the median score (120). A simple univariate analysis was performed to assess the association between all the covariates and college student happiness to obtain the significant covariates. We then employed simple logistic regression analysis to estimate the crude effect of the personal, familial, and social factors on student happiness. Finally, multiple logistic regression analysis was performed to evaluate the effects after adjustment for the covariates. The effect was presented as odds ratio (OR) and 95% confidence interval (CI) and a *p*-value less than 0.05 was considered to be statistically significant. The analysis reliability was 0.700, and the analytical validity was 0.958.

## 3. Results

Out of the 2186 respondents, 1094 participants (50.0%) reported happiness. The associations between potential factors and student happiness are shown in Table 1. The prevalence of happiness was higher in females (52.7%), those with standard BMI (51.9%), good health status (51.3%), frequent contact with family (55.0%), and good relations with their parents (54.5%), as well as those having high exam scores (63.7%), going to sleep early (before 24:00) (58.0%), and falling in love (55.4%).

The association between student happiness and their personal factors is shown in Table 2. The results found happiness was significantly associated with health status, with the adjusted OR (95% CI) = 3.41 (2.01–5.79) for healthy students compared to unhealthy students. In addition, we found that age was negatively associated with happiness, OR (95% CI) = 0.79 (0.63–0.98).

The association between student happiness and familial factors is presented in Table 3. The results show that both frequent contact with family and a harmonious relationship with parents were positively associated with happiness, with adjusted ORs (95% CIs) of 1.42 (1.17–1.71) and 2.32 (1.83–2.95), respectively.

Student happiness is associated with several social factors (Table 4). Firstly, a higher exam scores significantly enhanced student happiness with adjusted OR (95% CI) = 1.87 (1.51–2.32). Secondly, going to sleep earlier made students happier, the adjusted OR (95% CI) = 1.50 (1.24–1.81), as compared with going to sleep late. Thirdly, relationship status also affected student happiness, as the results found that students who were in a love relationship had significantly higher happiness than single students with adjusted OR = 1.32 (1.09–1.60).

## 4. Discussion

The nationwide cross-sectional study comprehensively explored factors affecting the happiness of college students in China. The results showed that happiness was significantly associated with a student’s personal, familial, and social factors. Students in good health, with high academic performance, who went to sleep earlier, and who were in a loving relationship, were happier. In addition, those who were in frequent contact with their families or who had a good relationship with their parents, were happier. Since university is a critical period for personal growth, these findings provide guidance for the public and the government to cultivate happiness in university students so as to help their future development.

The survey found that younger students are happier. This is consistent with the globally recognized U-shaped relationship between happiness and age, where happiness decreases with age before the middle age [62]. A cross-sectional survey of college students in Chile found that age had a negative association with happiness [40], and another study observed that first- and second-year pharmacy students had higher levels of happiness than third- and fourth-year students [63]. It also found that healthy students are happier. A cross-sectional study of the World Database on Happiness also shows a consistently positive association between health and happiness [64] and a study in Thailand found that healthy people are happier and less likely to feel lonely and hopeless [41].

The results suggested that contact with family and the relationship with parents were significantly associated with student happiness. This is consistent with the widely known attachment theory that individuals have to communicate meaningfully with each other to lead a happy life [65]. Support from family increases a student’s life satisfaction and provides them with additional emotional help and encouragement, which are conducive to student happiness. Studies have shown that the family environment has a big impact on an individual’s happiness [66]. A survey of medical students found that those who lived with their families reported significantly higher levels of happiness than those who lived in residence [38]. The effect that the relationship between parents has on childhood happiness or mental health, has been widely observed [45]. One study showed that parental conflict can lead to mental illness and lower happiness in Chinese children [67], while a national cohort study found that parental relationships are linked to children’s depressive symptoms in China [68]. A retrospective study has also shown that parental relationships are associated with a child’s psychological development, and this is even more important than paying direct attention to the child’s education [69].

In addition, it observed that doing well in exams had a positive association with happiness, which is consistent with previous research relating grades with happiness. Several cross-sectional studies have found a link between academics and happiness [42,70,71] and a longitudinal study suggested there may be a causal relationship between academics and happiness [72]. What’s more, a survey of college student happiness shows that happiness comes at least in part from academic achievement [73]. Nevertheless, some surveys found that happiness was not associated with college grade point average [74,75,76] and found that happiness has nothing to do with academic performance.

It found that going to sleep early had a significantly positive correlation with happiness. Numerous previous studies support this finding. A cross-sectional study of Japanese students discovered that staying up late was associated with being 1.45 times less happy than going to bed early [77]. Another study in Japan observed that the levels of subjective happiness were strongly associated with the prevalence of sleep problems, and a linear dose–response relationship was observed between sleep problems and subjective happiness scores [35]. Another cross-sectional study found that poor sleep efficiency was more prevalent among people who are less happy [78]. Moreover, an exploratory study using non-contact sheet sensors found that college student happiness was associated with taking a shorter time to fall asleep [79].

This study showed that students who are in a loving relationship are happier. Separated from their families, they transfer emotion to classmates, friends, and other peers. Once a love relationship is established, this intimate relationship may become an important source of emotional support for college students and improve their sense of identity and happiness. Studies have shown that non-single people are happier than single youth [38,80]. Researchers have also found that non-single college students reported higher levels of life satisfaction and happiness [38,81].

There are many studies on happiness, but the survey of college students is limited [35]. Further, there are few studies on China’s nationwide college students, such as some for a certain grade [60], some for a certain region [82], and more abroad [60]. This research was conducted in 2020, which investigated college students from all over China.

Some limitations should be acknowledged in this study. Firstly, some possible biases may be present in this study. Participant responses may confound our research because the study was based on self-administered questionnaires. Secondly, it used a cross-sectional survey, which is only able to find associations. Further research needs to use a prospective approach, which records the state of change over time through longitudinal studies to determine the causal relationships. Thirdly, this survey group included only Chinese students, so whether our conclusions are generally applicable remains to be considered. However, given a large number of students in our study and the fact that the happiness of Chinese students has been proved to be relatively stable [83], the study is still instructive. Moreover, we only considered part of the factors that influence college student happiness, and there are potential variables that are not included. Furthermore, the arrival of COVID-19 brought serious impacts to all walks of life, including people’s physical and mental health. In this study, the unexpected and inevitable lead to a deviation in the survey. However, the research is conducted in the form of an online questionnaire and considering that the sample was of college students, and communication tools such as mobile phones and computers are basically inseparable from students, the impact of the epidemic is not great. On the other hand, the study was carried out from January to June 2020. At this time, college students across the country had gone home because of China’s Spring Festival holiday, and because the epidemic had already caused a shutdown in the first month of the lunar calendar, college students stayed home during the survey period. Finally, and most importantly, the study has not considered the role of the built environment. The indoor, neighborhood, and city-level built environment have an important impact on the happiness of residents [84,85,86,87]. In addition, urban air pollution lowers Chinese urbanites’ happiness [88]. Therefore, future happiness research will add more potential variables and external environmental factors.

## 5. Conclusions

Happiness is more than a transient feeling in life but is also of great importance for the development and future careers of university students, and lasting happiness will especially promote a more productive, cohesive, caring, and sustainable society. This nationwide survey found that student happiness is significantly associated with their personal (health state), familial (contact with family and relation with parents), and social (academic performance, sleep habits, and love) factors. The findings indicate that the decline of student physical fitness levels, the reduced contact with family, and the increased use of electronic products (such as mobile phones and games) that contribute to going to sleep late and contribute to poor academic performance, may be the primary, underlying reasons for the decline in university student happiness in recent years. 

## Figures and Tables

**Figure 1 ijerph-19-04713-f001:**
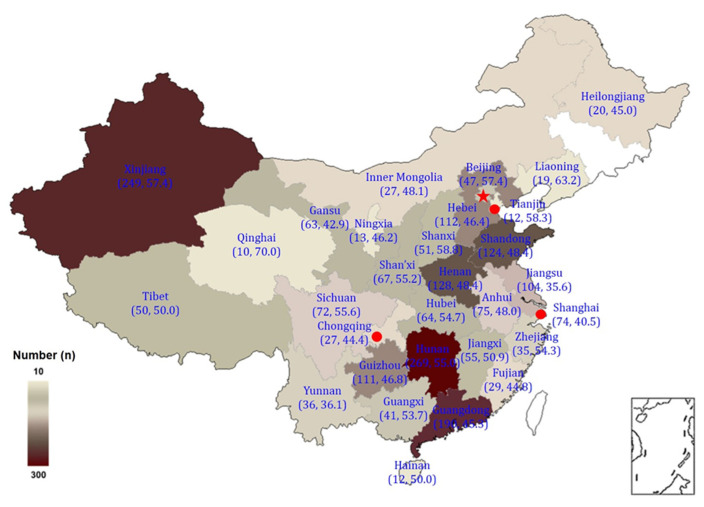
Distribution of the surveyed number, and happiness rate (*n*, %) among students from universities across China (*n* = 2186).

**Figure 2 ijerph-19-04713-f002:**
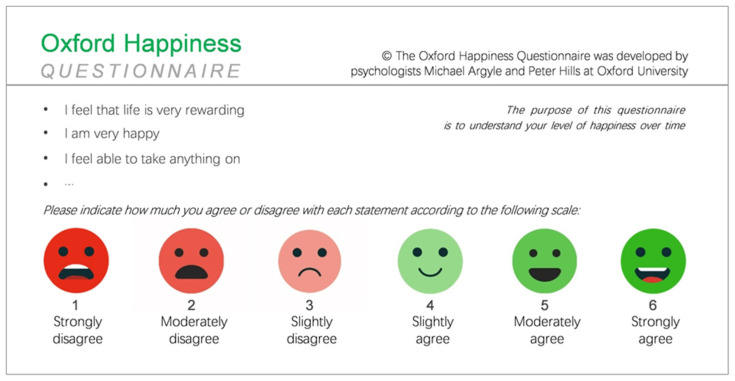
Oxford Happiness Questionnaire.

**Table 1 ijerph-19-04713-t001:** Covariate information for the students stratified by happiness in this study (*n* = 2186).

	Total		Happiness		
	Number *n*	(%)	Case *n*	(%)	*p*-Value
Total	2186	(100)	1094	(50.0)	—
**Personal factors**					
Sex					
Male	1130	(52)	538	(47.6)	**0.018**
Female	1056	(48)	556	(52.7)	
Age (years)					
≤21	1146	(52)	578	(50.4)	0.702
>21	1040	(48)	516	(49.6)	
Body mass index (BMI)					
Abnormal range (<18.5 or >23.9)	845	(39)	398	(47.1)	**0.029**
Normal range (18.5–23.9)	1341	(61)	696	(51.9)	
Grade					
Undergraduate	1682	(77)	826	(49.1)	0.109
Graduate	504	(23)	268	(53.2)	
Discipline					
Arts	221	(10)	105	(47.5)	0.454
Science and engineering	1567	(72)	780	(49.8)	
Medicine	398	(18)	209	(52.5)	
Health status					
Illness/Sick	91	(4)	19	(20.9)	<**0.001**
Well	2095	(96)	1075	(51.3)	
**Family factors**					
Contact with family					
Seldom	863	(40)	367	(42.5)	<**0.001**
Frequently	1323	(60)	727	(55.0)	
Relation with parents					
Not good	418	(19)	131	(31.3)	<**0.001**
Good	1768	(81)	963	(54.5)	
**Social factors**					
Exam scores					
Low	1695	(78)	781	(46.1)	<**0.001**
High	491	(23)	313	(63.7)	
Living expenses (RMB per month)					
≤2000	1755	(80)	872	(49.7)	0.498
>2000	431	(20)	222	(51.5)	
Sleeping habits					
Late (after 24:00)	1439	(66)	661	(45.9)	<**0.001**
Early (before 24:00)	747	(34)	433	(58.0)	
Relationship status					
Single	1457	(67)	690	(47.4)	<**0.001**
Being in love	729	(33)	404	(55.4)	

The values *p* < 0.05 were in bold.

**Table 2 ijerph-19-04713-t002:** Odds ratio (95%CI) of college student happiness for personal factors (*n* = 2186).

		Crude OR (95% CI)	Adjusted OR (95% CI) ^#^
Sex	Female vs. Male	1.22 (1.03–1.45) *	1.10 (0.91–1.33)
Age (years)	>21 vs. ≤21	0.97 (0.82–1.15)	0.79 (0.63–0.98) *
Body mass index	Normal vs. Abnormal	1.21 (1.02–1.44) *	1.15 (0.96–1.37)
Grade	Graduate vs. Undergraduate	1.18 (0.96–1.44)	1.14 (0.88–1.48)
Discipline	Arts	1.00	1.00
	Science and engineering	1.10 (0.83–1.45)	1.18 (0.86–1.61)
	Medicine	1.22 (0.88–1.70)	1.22 (0.85–1.73)
Health status	Well vs. Illness	3.99 (2.39–6.67) ***	3.41 (2.01–5.79) ***

^#^ Adjusted for all the covariates in Table 1. * *p* ≤ 0.05. *** *p* ≤ 0.001.

**Table 3 ijerph-19-04713-t003:** Odds ratio (95%CI) of college student happiness for family factors (*n* = 2186).

		Crude OR (95% CI)	Adjusted OR (95% CI) ^#^
Contact with family	Frequently vs. Seldom	1.65 (1.39–1.96) ***	1.42 (1.17–1.71) ***
Relation with parents	Good vs. Not good	2.62 (2.09–3.29) ***	2.32 (1.83–2.95) ***

^#^ Adjusted for all the covariates in Table 1. *** *p* ≤ 0.001.

**Table 4 ijerph-19-04713-t004:** Odds ratio (95%CI) of college student happiness for social factors (*n* = 2186).

		Crude OR (95% CI)	Adjusted OR (95% CI) ^#^
Exam scores	High vs. Low	2.06 (1.67–2.53) ***	1.87 (1.51–2.32) ***
Living expenses (RMB/month)	>2000 vs. ≤2000	1.08 (0.87–1.33)	0.93 (0.74–1.17)
Sleeping habit	Early vs. Late	1.62 (1.36–1.94) ***	1.50 (1.24–1.81) ***
Relationship status	Being in love vs. Single	1.38 (1.16–1.65) ***	1.32 (1.09–1.60) **

^#^ Adjusted for all the covariates in Table 1. ** *p* ≤ 0.01. *** *p* ≤ 0.001.

## Data Availability

The data presented in this study are available on request from the corresponding author.

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
