# Peer review of "Happiness in University Students: Personal, Familial, and Social Factors: A Cross-Sectional Questionnaire Survey"

_ijerph, 2022, doi:10.3390/ijerph19084713_

Round 1

Reviewer 1 Report

This is a good start. However, I would like to see the argument build for the specific variables in the analysis. The literature review feels more like a laundry list of factors / studies that have examined different variables that have been predictors of happiness. It's not a strong argument being built that lead to the conclusion that those factors are the most important factors that predict happiness. Also, it would be important to write about the construct of happiness itself. It's necessary and needed. There is no real definition of happiness presented. How is the construct defined by you?

The generalized statement that happiness levels among university students are low needs to be cited. Low where? 

A study cited in 2013 is hardly a recent study - anything newer? (line 43).  

Why is the main variable starting as a continuous variable and you convert it into a categorical variable? That doesn't make senses.  Happiness, indeed, is a variable with different degrees and making it dichotomous decreases the accuracy of the variable itself. 

Author Response

This is a good start. However, I would like to see the argument build for the specific variables in the analysis. The literature review feels more like a laundry list of factors/studies that have examined different variables that have been predictors of happiness. It's not a strong argument being built that lead to the conclusion that those factors are the most important factors that predict happiness. Also, it would be important to write about the construct of happiness itself. It's necessary and needed. There is no real definition of happiness presented. How is the construct defined by you?

Response:

Thank you very much for your positive and encouraging comments on our manuscript. We have revised the manuscript according to your suggestions and hope the revised manuscript would be satisfied.

This study mainly looks for the factors that may affect college students' well-being by consulting relevant literature. Of course, this is not comprehensive, so in the part of Discussion, a new article is added: Moreover, we only considered part of the factors that influence college students' happiness, and there are potential variables that are not included.

As for the definition of happiness, the following contents are added in the Introduction: Many studies have put forward their own views on the definition of happiness. For example, Veenhoven (1992) indicted that: "Happiness is the degree to which a person judges whether his quality of life is satisfactory." In a word, happiness is a subjective and internal emotion and cognition.

The generalized statement that happiness levels among university students are low needs to be cited. Low where?

Response:

Thanks for your suggestions.

In the Introduction, the data results of low happiness level of college students are added: A 2017 survey showed that adults aged 18-25 had the highest rate of depression at 13.1 percent, compared with 7.7 percent for those aged 26-49 and 4.7 percent for those aged 50 and above. In addition, the incidence of suicidal thoughts in young people aged 18-25 was twice as high as in other age groups, and the incidence of suicide was 4.5 times higher than in other age groups. And According to the 2018 National College Health Assessment Survey, 13 percent of college students have suicidal thoughts and about 2 percent have attempted suicide at least once in the past year.

A study cited in 2013 is hardly a recent study - anything newer? (line 43).

Response:

Thanks for your suggestion. A 2017 study was added to the Introduction.

Why is the main variable starting as a continuous variable and you convert it into a categorical variable? That doesn't make senses.  Happiness, indeed, is a variable with different degrees and making it dichotomous decreases the accuracy of the variable itself.

Response:

Thanks for your suggestions.

Through literature inquiry, the results of Oxford well-being questionnaire can be processed by binary variables. In this study, it is mainly to explore the factors related to college students' well-being, and the processing of binary variables can be more clearly expressed.

Reviewer 2 Report

Thank you for submitting the manuscript to this journal. You have chosen an interesting subject to do this study. Some issues need to be clarified. I have summarized these concerns below: the following comments need to be taken with care in order to improve the quality of the manuscript for publishing.

TITLE

In the title it would be appropriate for the authors to indicate the design of the study.

ABSTRACT

The elaboration of the abstract does not adapt to the recommendations of the journal, the subsections (Objectives, methods...) must be eliminated. The authors use the first person plural, change to impersonal forms.

KEYWORDS

keywords must be in alphabetical order.

INTRODUCTION

Authors need to adjust references to the style of the journal. References must appear numbered and []. Line 43, indicate a recent study and the article is from 2013, almost 10 years old. Student happiness has been explored and there are numerous more recent studies that could be incorporated into the introduction.

Line 51-53, the authors do not provide references for the information. It is important that the entire introduction is referenced and based on the most current references possible.

Line 54 The origins of happiness are very complicated (Easterlin 2003) and then address the factors. It is necessary to review the introduction and to address the topic from the general to the specific, and enter the university. It would be interesting to include a paragraph indicating the importance of exploring the topic.

Line 87-93 There have been calls on governments and organizations at all levels 87 throughout the world to carry out campaigns to improve students’ happiness. In re-88 sponse to this call, we conducted a nationwide survey of happiness in university students 89 in China during 2020. The strength of our study lies in the comprehensive consideration 90 of the influence of multi-level factors, from individual, through familial to social factors 91 affecting the happiness of college students. Our findings provide novel suggestions for 92 the government and the public to cultivate happiness in university students. This information does not apply in the introduction section.

The authors do not indicate the objective of the study at the end of the introduction.

METHODS

protocol was approved by the Ethics Committee of Central South University, please, could you indicate the assigned number.

Review the wording and avoid the use of the first person plural.

This section is incomplete the following subsections are missing:

Design and Participant (indicate criteria for inclusion and exclusion of participants, calculation of sample size...), guide used based on the design to guarantee quality.

The questionnaires were collected online? they need to provide more information on data collection.

Questionnaire survey

They could indicate the reliability and validity of the questionnaire used in that population.

RESULTS

Check the wording of the results, information from the text is duplicated in the tables.

DISCUSSION

Avoid the first person plural in the wording.

The authors do not include the limitations of the study.

It would be necessary to include new future lines to investigate, after the completion of the study.

It would be convenient to indicate the implications of the article.

The manuscripts with which the results of the studies are contracted are very old, it would be necessary to update the references.

CONCLUSION

The conclusions must be clear, direct and respond to the stated objective, for this reason the authors must review them. Eliminate the references that they include, and base their writing on their study.

REFERENCES

References must be ordered and numbered according to their appearance in the text.

Author Response

Thank you for submitting the manuscript to this journal. You have chosen an interesting subject to do this study. Some issues need to be clarified. I have summarized these concerns below: the following comments need to be taken with care in order to improve the quality of the manuscript for publishing.

Response:

Thank you very much for your positive and encouraging comments on our manuscript. We have revised the manuscript according to your suggestions and hope the revised manuscript would be satisfied.

TITLE

In the title it would be appropriate for the authors to indicate the design of the study.

Response:

Thanks for your suggestions.

The Title has been changed to Happiness in university students: Personal, familial and social factors: A cross-sectional questionnaire survey.

ABSTRACT

The elaboration of the abstract does not adapt to the recommendations of the journal, the subsections (Objectives, methods...) must be eliminated. The authors use the first person plural, change to impersonal forms.

Response:

Thanks for your suggestions.

See the revised manuscript for details.

KEYWORDS

keywords must be in alphabetical order.

Response:

Thanks for your suggestions.

See the revised manuscript for details.

INTRODUCTION

Authors need to adjust references to the style of the journal. References must appear numbered and []. Line 43, indicate a recent study and the article is from 2013, almost 10 years old. Student happiness has been explored and there are numerous more recent studies that could be incorporated into the introduction.

Response:

Thanks for your suggestions.

The format of references has been adjusted.

A 2017 study was added to the Introduction.

See the revised manuscript for details.

Line 51-53, the authors do not provide references for the information. It is important that the entire introduction is referenced and based on the most current references possible.

Response:

Thanks for your suggestions.

The statistical results of the number of students published in 2021 are for reference.

See the revised manuscript for details.

Line 54 The origins of happiness are very complicated (Easterlin 2003) and then address the factors. It is necessary to review the introduction and to address the topic from the general to the specific, and enter the university. It would be interesting to include a paragraph indicating the importance of exploring the topic.

Response:

Thanks for your suggestions.

See the revised manuscript for details.

Line 87-93 There have been calls on governments and organizations at all levels 87 throughout the world to carry out campaigns to improve students’ happiness. In re-88 sponse to this call, we conducted a nationwide survey of happiness in university students 89 in China during 2020. The strength of our study lies in the comprehensive consideration 90 of the influence of multi-level factors, from individual, through familial to social factors 91 affecting the happiness of college students. Our findings provide novel suggestions for 92 the government and the public to cultivate happiness in university students. This information does not apply in the introduction section.

Response:

Thanks for your suggestions.

See the revised manuscript for details.

The authors do not indicate the objective of the study at the end of the introduction.

Response:

Thanks for your suggestions.

At the end of the Introduction, the following contents are added: So the purpose of this study is to explore the main factors influencing students' happiness, in order to find useful suggestions to improve the quality of their development, and it is beneficial for society and country as well.

METHODS

protocol was approved by the Ethics Committee of Central South University, please, could you indicate the assigned number.

Response:

Thanks for your suggestions.

The assigned number is XYGW-2020-17.

Review the wording and avoid the use of the first person plural.

Response:

Thanks for your suggestions.

See the revised manuscript for details.

This section is incomplete the following subsections are missing:

Design and Participant (indicate criteria for inclusion and exclusion of participants, calculation of sample size...), guide used based on the design to guarantee quality.

Response:

Thanks for your suggestions.

Collect data in the form of online questionnaire. In the questionnaire, there is information about the educational background of the person filling in the questionnaire, which can be used to screen the effective questionnaire. In addition, in this year's one-step sorting, the abnormal height and weight information questionnaire and the regional questionnaire with too small sample size are excluded. The reliability of questionnaire analysis is increased through the above methods.

The questionnaires were collected online? they need to provide more information on data collection.

Response:

Thanks for your suggestions.

Collect data in the form of online questionnaire. See the revised manuscript for the collection and screening of data for details.

Questionnaire survey

Response:

Thanks for your suggestions.

See attached Table S1 for the contents of the questionnaire.

They could indicate the reliability and validity of the questionnaire used in that population.

Response:

Thanks for your suggestions.

The analysis reliability was 0.700, and the analytical validity was 0.958.

See the revised manuscript for details.

RESULTS

Check the wording of the results, information from the text is duplicated in the tables.

Response:

Thanks for your suggestions.

See the revised manuscript for details.

DISCUSSION

Avoid the first person plural in the wording.

Response:

Thanks for your suggestions.

See the revised manuscript for details.

The authors do not include the limitations of the study.

Response:

Thanks for your suggestions.

See the Discussion section of the revised manuscript for details.

It would be necessary to include new future lines to investigate, after the completion of the study.

Response:

Thanks for your suggestions.

See the revised manuscript for details.

It would be convenient to indicate the implications of the article.

Response:

Thanks for your suggestions.

See the first paragraph of the Discussion part of the revised manuscript for details.

The manuscripts with which the results of the studies are contracted are very old, it would be necessary to update the references.

Response:

Thanks for your suggestions.

See the revised manuscript for details.

CONCLUSION

The conclusions must be clear, direct and respond to the stated objective, for this reason the authors must review them. Eliminate the references that they include, and base their writing on their study.

Response:

Thanks for your suggestions.

See the revised manuscript for details.

REFERENCES

References must be ordered and numbered according to their appearance in the text.

Response:

Thanks for your suggestions.

See the revised manuscript for details.

Reviewer 3 Report

Happiness is a very important topic. This paper tries to analyse happiness in a student sample in China, using the Oxford Happiness Questionnaire. The variables studied include personal, familial and social information. Happiness was found to be associated with health, and decreased with age. In addition, happiness was positively related to frequent contact with family and a harmonious relationship, and with academic performance.

The authors need to justify the choice of the Oxford survey, as several other options exist to study happiness and life satisfaction, for instance the Satisfaction with Life Scale (SWLS). In fact some of the findings are very much in line with the results from studies of life satisfaction (see in particular the work by Ed Diener).

A limitation of the study is that it lacks novelty and some methodological choices are unclear. The sample is large and from several regions of China but how were these regions selected and how was the sample analysed for representativeness of the student population?

One final problem, but a very important one, is that the data were collected in 2020. It is not possible to conduct a large study in the population in 2020 without some consideration of the impact of the COVID19 pandemic.

Author Response

The authors need to justify the choice of the Oxford survey, as several other options exist to study happiness and life satisfaction, for instance the Satisfaction with Life Scale (SWLS). In fact some of the findings are very much in line with the results from studies of life satisfaction (see in particular the work by Ed Diener).

Response:

Thanks for your suggestions.

These items can easily be included in larger questionnaires in random order, and the opportunity is also used to reverse about half of the items. These changes should reduce the likelihood of contextual and adaptive responses.

A limitation of the study is that it lacks novelty and some methodological choices are unclear. The sample is large and from several regions of China but how were these regions selected and how was the sample analysed for representativeness of the student population?

Response:

Thanks for your suggestions.

When sampling, it is carried out in a random way. When sorting, those questionnaires with small regional sample size will be eliminated to increase the reliability of analysis.

One final problem, but a very important one, is that the data were collected in 2020. It is not possible to conduct a large study in the population in 2020 without some consideration of the impact of the COVID19 pandemic.

Response:

Thanks for your suggestions.

The study was carried out in the form of online questionnaire, so a wide range of data and information were collected during the epidemic.

Round 2

Reviewer 2 Report

All the recommendations were addressed. 

Author Response

Thank you very much for the professional comments which make our manuscript much better.

Reviewer 3 Report

I read the replies to the reviewers and they do not seem to address the comments at all. The authors need to be more careful in how they consider these comments, whether they agree or not. The replies have to be clear and very specific on each point.

Round 3

Reviewer 3 Report

The authors have addressed the comments of the reviewers